


# New-generation NASA Aura Ozone Monitoring Instrument (OMI) volcanic SO₂ dataset: Algorithm description, initial results, and continuation with the Suomi-NPP Ozone Mapping and Profiler Suite (OMPS)

Can Li[1,2], Nickolay A. Krotkov[2], Simon Carn[3], Yan Zhang[1,2], Robert J. D. Spurr[4], Joanna Joiner[2]

[1]Earth System Science Interdisciplinary Center, University of Maryland, College Park, MD 20742. USA
[2]NASA Goddard Space Flight Center, Greenbelt, MD 20771, USA
[3]Department of Geological and Mining Engineering and Sciences, Michigan Technological University, Houghton, MI 49931, USA
[4]RT Solutions, Inc., Cambridge, MA 02138, USA

*Correspondence to*: Can Li (can.li@nasa.gov)

**Abstract.** Since the fall of 2004, the Ozone Monitoring Instrument (OMI) has been providing global monitoring of volcanic SO₂ emissions, helping to understand their climate impacts and to mitigate aviation hazards. Here we introduce a new generation OMI volcanic SO₂ dataset based on a principal component analysis (PCA) retrieval technique. To reduce retrieval noise and artifacts as seen in the current operational linear fit (LF) algorithm, the new algorithm, OMSO2VOLCANO, uses characteristic features extracted directly from OMI radiances in the spectral fitting, thereby helping to minimize interferences from various geophysical processes (e.g., O₃ absorption) and measurement details (e.g., wavelength shift). To solve the problem of low bias for large SO₂ total columns in the LF product, the OMSO2VOLCANO algorithm employs a table lookup approach to estimate SO₂ Jacobians (i.e., the instrument sensitivity to a perturbation in the SO₂ column amount) and iteratively adjusts the spectral fitting window to exclude shorter wavelengths where the SO₂ absorption signals are saturated. To first order, the effects of clouds and aerosols are accounted for using a simple Lambertian equivalent reflectivity approach. As with the LF algorithm, OMSO2VOLCANO provides total column retrievals based on a set of pre-defined SO₂ profiles from the lower troposphere to the lower stratosphere, including a new profile peaked at 13 km for plumes in the upper troposphere. Examples given in this study indicate that the new dataset shows significant improvement over the LF product, with at least 50% reduction in retrieval noise over the remote Pacific. For large eruptions such as Kasatochi in 2008 (~1700 kt total SO₂) and Sierra Negra in 2005 (>1100 DU maximal SO₂), OMSO2VOLCANO generally agrees well with other algorithms that also utilize the full spectral content of satellite measurements, while the LF algorithm tends to underestimate SO₂. We also demonstrate that, despite the coarser spatial and spectral resolution of the Suomi National Polar-orbiting Partnership (Suomi-NPP) Ozone Mapping and Profiler Suite (OMPS) instrument, application of the new PCA algorithm to OMPS data produces highly consistent retrievals between OMI and OMPS. The new PCA algorithm is therefore capable of continuing the volcanic SO₂ data record well into the future using current and future hyperspectral UV satellite instruments.





## 1 Introduction

Volcanic emissions, while collectively a smaller source of sulfur dioxide ($SO_2$) than anthropogenic emissions (e.g., Bluth et al., 1993), are a dominant and highly variable natural forcing to the Earth's climate system. Explosive eruptions such as El Chichón in 1982 (Krueger, 1983) and Mt. Pinatubo in 1991 (McCormick et al., 1995 and references therein) directly inject

large amounts of $SO_2$ and other species into the stratosphere, producing secondary sulfate aerosols that can remain at high altitudes for years. Such large eruptions are rare, but they have been found to cause significant perturbations to global surface temperature and atmospheric circulation patterns (e.g., Robock and Mao, 1995, Robock et al., 2007; Stenchikov et al., 2002). Recent studies (Ridley et al., 2014; Solomon et al., 2011; Vernier et al., 2011) suggest that more frequent, moderate eruptions may also be a far more important source of stratospheric aerosols than previously thought, and may have

contributed to the slower recent rate of global warming (Santer et al., 2014). $SO_2$ emitted by passive volcanic degassing is normally too short-lived to reach the stratosphere, but its regional climate impact can be significant through the interaction between secondary sulfate aerosols and clouds (Schmidt et al., 2012; Yuan et al., 2011). In addition to their climate effects, volcanic plumes also pose severe threats to health and human lives and aviation safety (e.g., Carn et al., 2009; Stohl et al., 2011). To better understand volcanic climate forcing, it is important to acquire accurate estimates of emissions from various

types of volcanic activity. Mitigating volcanic hazards, on the other hand, requires global monitoring of volcanic plumes on a timely basis.

Since the first demonstration by Krueger (1983), satellite retrievals of volcanic $SO_2$ using ultraviolet (UV) instruments have become a critical tool in studies of volcanism as well as in management of aviation safety. The satellite volcanic $SO_2$ data record dates back to the 1970s (e.g., Carn et al., 2016). However, earlier measurements (pre-1990s) were

generally limited to larger eruptions (e.g., Krueger et al., 1995; McPeters, 1993), due the small number of discrete wavelengths and relatively large footprint size characteristic of heritage instruments such as the Total Ozone Mapping Spectrometer (TOMS). Since the 1990s, hyperspectral UV instruments have made measurements at hundreds of wavelengths, allowing $SO_2$ absorption features to be more clearly separated from interfering processes. This has been demonstrated with the Global Ozone Monitoring Experiment (GOME, Burrows et al., 1999; Eisinger and Burrows, 1998),

GOME-2 (Nowlan et al., 2011; Rix et al., 2009), and the SCanning Imaging Absorption spectroMeter for Atmospheric CHartographY (SCIAMACHY, Lee et al., 2008).

Among hyperspectral instruments that are currently operating in orbit, the Dutch-Finnish Ozone Monitoring Instrument (OMI, Levelt et al., 2006), launched on board NASA's Aura spacecraft in 2004, offers the best ground resolution ($13 \times 24$ km$^2$ at nadir), along with a wide spectral range (270-500 nm) and contiguous daily global coverage - features that

make it an effective instrument for global $SO_2$ monitoring. In contrast to TOMS, OMI permits the detection of weak $SO_2$ emissions such as those from coal-fired power plants or quiescent degassing volcanoes (e.g., Carn et al., 2008; Hsu et al., 2012; Li et al., 2010a, 2010b; Lu et al., 2010), even with the at-launch algorithms (Krotkov et al., 2006, 2008; Yang et al., 2007) that utilize only a small fraction of the available wavelengths. The linear fit (LF) algorithm (Yang et al., 2007) is





presently the operational algorithm for NASA's standard OMI volcanic $SO_2$ product. It uses ten OMI wavelengths between 310.8 nm and 360.2 nm, and produces estimates of the total $SO_2$ vertical column density (VCD) assuming three different prescribed $SO_2$ profiles with center of mass altitudes (CMAs) at 3 km (lower troposphere, TRL), 8 km (middle troposphere, TRM), and 18 km (lower stratosphere, STL). The presumed CMAs were selected to represent $SO_2$ from passively degassing

volcanoes, moderate eruptions, and large explosive eruptions, respectively. In addition to the OMI standard product, the LF algorithm has also been implemented to produce $SO_2$ retrievals quickly from direct readout data, as well as near real-time $SO_2$ data from OMI and the Ozone Mapping and Profiler Suite (OMPS) nadir mapper aboard the NASA/NOAA Suomi-National Polar-orbing Partnership (Suomi-NPP) spacecraft. The LF algorithm has been widely used in studies of volcanic $SO_2$ and aviation safety applications, but it has a number of limitations. For example, it is known to significantly

underestimate $SO_2$ VCDs in relatively large eruptions, due to signal saturation at shorter wavelengths (e.g., Krotkov et al., 2010; Yang et al., 2009). The algorithm also has large biases over clean background areas, making it difficult to track volcanic plumes from relatively small eruptions.

Several studies have demonstrated that OMI $SO_2$ retrievals can be improved by exploiting the full spectral content of the hyperspectral OMI measurements. Theys et al. (2015) produced an OMI $SO_2$ product with reduced noise and bias

using a DOAS (Differential Optical Absorption Spectroscopy) scheme. For large volcanic signals, they used longer wavelengths (up to 360-390 nm) where $SO_2$ is only weakly absorbing to avoid signal saturation. The iterative spectral fitting (ISF, and its variant, direct spectral fitting or DSF) algorithm developed by Yang et al. (2009) also utilized OMI spectral measurements at longer wavelengths and produced much greater $SO_2$ VCDs for $SO_2$-rich eruptions such as Sierra Negra in 2005 (Yang et al., 2009) and Kasatochi in 2008 (Krotkov et al., 2010) than the corresponding VCDs derived from the LF

algorithm. However, the ISF/DSF algorithm relies on computationally intensive online radiative transfer calculations and instrument-specific soft calibration, hindering its operational implementation.

Recently, we introduced a retrieval technique based on principal component analysis (PCA) of satellite-measured radiances (Li et al., 2013; 2015). Unlike DOAS or ISF/DSF, the PCA technique is a data-driven approach and uses a set of principal components (PCs) extracted directly from satellite radiance data in the spectral fitting. It benefits from the fact that

the PCs that account for the most spectral variance have characteristics associated with various geophysical processes (e.g., $O_3$ absorption, rotational Raman scattering (RRS)) or measurement details (e.g., wavelength shift, variation in dark current). This allows these various interferences in $SO_2$ retrievals to be minimized without extensive forward calculation or explicit instrument characterization, leading to an efficient implementation with relatively small biases and noise in the retrieved $SO_2$ VCDs.

Our PCA algorithm has been implemented for operational production of the new generation NASA standard OMI planetary boundary layer (PBL) dataset (used for air quality studies). As compared with the previous OMI PBL $SO_2$ dataset (Kroktov et al., 2006), the new product improves the detection limit for point anthropogenic sources by a factor of two (Fioletov et al., 2015; 2016), enabling the detection of a large number of sources missing in current emission inventories (McLinden et al., 2016) as well as regional trends in $SO_2$ pollution (e.g., Krotkov et al., 2016). In this paper, we extend the





PCA algorithm to volcanic retrievals and introduce the new generation NASA OMI volcanic $SO_2$ product. We describe the algorithm in section 2, and present results for selected scenarios in section 3, including background regions and several volcanic eruptions; the focus is on comparisons with retrievals from the current operational LF algorithm. Another advantage of the PCA approach is its ability to generate consistent retrievals between different instruments, and in section 4 we discuss

the prospect of continuing the long-term OMI data record with Suomi-NPP OMPS.

## 2 Algorithm description

Our new OMI volcanic $SO_2$ algorithm (hereafter referred to as OMSO2VOLCANO) comprises two main components: the first step (enclosed by the dashed blue box in the algorithm flowchart; Figure 1) is designed to identify pixels having strong $SO_2$ signals and provide initial estimates of $SO_2$ VCD ($\Omega_{SO2}$). The second step produces more accurate estimates of volcanic

$SO_2$ VCD through iterative spectral fitting.

### 2.1 Step 1: PCA and initial fit

The first step of the OMSO2VOLCANO algorithm is essentially identical to the operational OMI PBL $SO_2$ algorithm that has been previously described in detail elsewhere (Li et al., 2013) and is only briefly reviewed here. In short, for each OMI orbit, we process the 60 rows (cross-track positions) one at a time, employing a PCA technique to extract principal

components (PCs or $v_i$) for the spectral range 310.5-340 nm from the sun-normalized radiance spectra of ~1000 pixels along the flight direction (after excluding pixels with high slant column ozone). The PCs are ranked in descending order according to the spectral variance they each explain. If derived from $SO_2$-free regions, the first several PCs that account for the most of the variance are representative of geophysical processes unrelated to $SO_2$ such as ozone absorption and RRS, as well as measurement details such as wavelength shift. We then obtain initial estimates of $SO_2$ VCD ($\Omega_{SO2\_ini}$) and the coefficients of

the PCs ($\boldsymbol{\omega}$) by fitting the first $n_v$ (non-$SO_2$) PCs and the $SO_2$ Jacobians ($\partial N/\partial \Omega_{SO_2}$) to the measured radiance spectrum (in this case the quantity $N$, which is the logarithm of the sun-normalized radiances, $I$):

$$N\left(\omega, \Omega_{SO_2}\right) = \sum_{i=1}^{n_v} \omega_i v_i + \Omega_{SO_2} \frac{\partial N}{\partial \Omega_{SO_2}}. \tag{1}$$

The $SO_2$ Jacobians represent the sensitivity of Sun-normalized earthshine radiances ($I$ or its logarithm, $N$) at the top of the atmosphere (TOA) to a unit perturbation in $\Omega_{SO2}$, and were pre-calculated with the vector radiative transfer code

VLIDORT (Spurr, 2008). As with the PBL $SO_2$ algorithm (Li et al., 2013), in this part of the algorithm we use a fixed $SO_2$ Jacobian spectrum in Eq. (1), calculated assuming that $SO_2$ is predominantly in the lowest 1000 m of the atmosphere and that the observation is made under cloud-free conditions with fixed surface albedo (0.05), surface pressure (1013.25 hPa), solar zenith angle (30°), viewing zenith angle (0°), and pre-set $O_3$ and temperature profiles (with $O_3$ VCD = 325 DU).

In most cases, we use $n_v$ = 20 PCs in the fitting. However, in the presence of strong $SO_2$ signals (e.g., a volcanic

plume), the inclusion of that many PCs in Eq. (1) can introduce collinearity, as some of the leading PCs may contain an $SO_2$



absorption signature. To avoid this, we examine the first 20 PCs and only use $n_v = i-1$ PCs if the $i^{th}$ PC is found to be significantly correlated with $SO_2$ cross sections. We then exclude pixels with large $\Omega_{SO2\_ini}$ (outside of ±1.5 standard deviations for the row) and repeat the PCA and spectral fitting to get updated PCs and the $SO_2$ VCD. The output PCs and VCD are used as input to the second step of the OMSO2VOLCANO algorithm (see Figure 1).

**2.2 Step 2: Volcanic $SO_2$ Jacobians lookup table**

A main function of the second step of the OMSO2VOLCANO algorithm is to determine $SO_2$ Jacobians under various actual observation conditions. In our spectral range of interest (~310 to 340 nm), both $I$ (i.e., Sun-normalized earthshine radiances) and the $SO_2$ Jacobians depend on a number of factors including satellite geometry (solar zenith angle, SZA or $\theta_0$, viewing zenith angle, VZA or $\theta$, and relative azimuth angle, RAZ or $\phi$), surface reflectivity and pressure, cloud fraction and height,

aerosols, and the amount and vertical distribution of absorbing gases ($O_3$ and $SO_2$). However, for an operational global retrieval algorithm, it would be computationally too expensive to perform online radiative transfer calculations at many wavelengths to explicitly account for all of these factors. Moreover, some of these factors, such as the height of the volcanic $SO_2$ plume or the composition and size distribution of aerosols, are usually not well known or well defined at the time of retrieval. To simplify the problem and save computational expense, we have constructed a number of pre-computed lookup

tables for volcanic $SO_2$ Jacobians, following an approach similar to that used in TOMS and OMI total column $O_3$ retrievals (Bhartia and Wellemeyer, 2002). We assume that, to first order, the combined effects of clouds/aerosols/surface reflectivity on $SO_2$ Jacobians are accounted for with a simple Lambertian equivalent reflectivity (SLER or $R$) derived at the surface (Ahmad et al., 2004). We also neglect the effects of inelastic RRS on $SO_2$ Jacobians (RRS contribution to radiances is accounted for with PCs in Eq. (1)). On the basis of these assumptions, the backscattered radiances at TOA ($I$) for Rayleigh

multiple scattering can be calculated with the following equation:

$$I = I_0(\theta_0, \theta) + I_1(\theta_0, \theta) \cos \phi + I_2(\theta_0, \theta) \cos 2\phi + \frac{RI_r(\theta_0, \theta)}{(1-RS_b)}. \tag{2}$$

Together, the first three terms ($I_0$, $I_1$, and $I_2$) on the right-hand side of Eq. (2) represent the atmospheric contribution

to the radiances. The last term represents the surface contribution, where $RI_r$ represents the TOA radiance that is reflected once from the surface and transmitted through the atmosphere, and $(1 - RS_b)$ accounts for the effects of multiple reflections between the surface and the atmosphere, with $S_b$ being the fraction of surface-reflected radiation that is scattered back to the surface by the atmosphere. Note that these terms also depend on absorbing gases such as $O_3$ and $SO_2$, which are omitted from Eq. (2) for brevity. Taking the partial derivative with respect to the $SO_2$ VCD ($\Omega_{SO2}$), we obtain the following equation

for the calculation of $SO_2$ Jacobians:

$$\frac{\partial I}{\partial \Omega_{so2}} = \frac{\partial I_0(\theta_0, \theta)}{\partial \Omega_{so2}} + \frac{\partial I_1(\theta_0, \theta)}{\partial \Omega_{so2}} \cos \phi + \frac{\partial I_2(\theta_0, \theta)}{\partial \Omega_{so2}} \cos 2\phi + \frac{R}{(1-RS_b)} \frac{\partial I_r(\theta_0, \theta)}{\partial \Omega_{so2}} + \frac{R^2 I_r(\theta_0, \theta)}{(1-RS_b)^2} \frac{\partial S_b}{\partial \Omega_{so2}}. \tag{3}$$





To account for $O_3$ vertical distributions, we generated a set of $SO_2$ Jacobian lookup tables, one for each of the 21 standard $O_3$ climatology profiles used in OMI total $O_3$ retrievals (each profile represents an $O_3$ node), employing VLIDORT (Spurr, 2008) to compute the components in Eq. (3) ($I_0$, $I_1$, $I_2$, $I_r$, $S_b$, and their derivatives with respect to $\Omega_{SO2}$) for eight SZAs

(0-81°), eight VZAs (0-80°), and 15 $SO_2$ nodes (0-1000 DU) for the spectral range 311-342 nm at 0.05 nm resolution (*cf.* Table 1 for a list of the nodes). This was done for four prescribed Gaussian vertical $SO_2$ profiles with a full width at half maximum (FWHM) of ~2.3 km and different CMAs. Three of these profiles have the same plume CMAs as those used in the LF algorithm (3 km or TRL, 8 km or TRM, and 18 km or STL). Retrievals produced using these profiles are distributed as part of the new operational OMI $SO_2$ dataset (OMSO2 V1.3.0). The fourth $SO_2$ profile, centered at 13 km altitude, was

introduced for the generation of a new TRU (upper troposphere) $SO_2$ research product. Several eruptions during the OMI era, including Kasatochi in 2008, injected $SO_2$ to altitudes between 8 and 18 km, and the TRU retrievals should allow for more accurate retrievals for those eruptions.

**2.3 Step 2: Ancillary retrieval parameters, data processing, and fitting windows**

For a given pixel with SZA = $\theta_0$, VZA = $\theta$, and RAZ = $\phi$, a number of ancillary retrieval parameters are generated at

intermediate steps in the OMSO2VOLCANO algorithm, before the volcanic $SO_2$ Jacobians for the pixel can be determined. Two of these parameters, namely, the initial estimate of $SO_2$ VCD ($\Omega_{SO2\_ini}$) and an estimate of $O_3$ VCD ($\Omega_{O3}$), come from the first stage of the OMSO2VOLCANO algorithm (blue box in Figure 1). $\Omega_{O3}$ is from the operational OMI total column $O_3$ product (OMTO3) for most pixels, but for pixels having large $SO_2$ signals ($\Omega_{SO2\_ini}$ > 5 DU), it is interpolated from neighbouring pixels with small $\Omega_{SO2\_ini}$ in order to avoid overestimation of $\Omega_{O3}$ due to $SO_2$ contamination. The SLER ($R$) is

determined by first matching the measured and calculated radiances in Eq. (2) to derive $R$ at 342.5, 354.1, and 367.04 nm, where contributions from gaseous absorption and RRS processes are minimal, and then fitting a second-degree polynomial function to the three wavelengths to extrapolate $R$ to shorter wavelengths. The interpolation implicitly accounts for the combined effects of aerosols, clouds, and the surface on the spectral dependence of TOA radiances.

For each presumed $SO_2$ profile, the algorithm then selects two pre-computed lookup tables based on the $O_3$ VCD

and the latitude of the pixel, with the two $O_3$ nodes bracketing the input $\Omega_{O3}$ for the pixel ($\Omega_{node1}$ < $\Omega_{O3}$ < $\Omega_{node2}$). For the lookup table with $O_3$ VCD = $\Omega_{node1}$, a total of eight $SO_2$ Jacobian spectra are calculated using $\phi$ and the derived $R$ as input to Eq. (3) for two $SO_2$, two SZA, and two VZA nodes that bracket $\Omega_{SO2\_ini}$, $\theta_0$, and $\theta$, respectively. Next, the eight $SO_2$ Jacobian spectra are interpolated in two steps, a 2-D linear interpolation with respect to the cosines of the angles in the SZA-dimension and VZA-dimension followed by a 1-D linear interpolation between the two $SO_2$ nodes, leading to an estimated

$SO_2$ Jacobian spectrum for SZA = $\theta_0$, VZA = $\theta$, RAZ = $\phi$, $SO_2$ VCD = $\Omega_{SO2\_ini}$ and $O_3$ VCD = $\Omega_{node1}$. We repeat the above steps for the lookup table with $O_3$ VCD = $\Omega_{node2}$ to obtain another estimated $SO_2$ Jacobian spectrum for the same input, but for a different $O_3$ profile. A final interpolation between the two $O_3$ nodes is then performed to generate an estimated $SO_2$ Jacobian spectrum for the given pixel.





The SO$_2$ Jacobian spectrum is convolved with the OMI slit function, and then used along with the PCs from the first part of the algorithm (Figure 1) in least squares fitting to produce an updated estimate of SO$_2$ VCD ($\Omega_{SO2\_step1}$) for the pixel, which is then compared with $\Omega_{SO2\_ini}$. If | $\Omega_{SO2\_ini}$ - $\Omega_{SO2\_step1}$ | is greater than 0.1 DU or 1% for pixels with SO$_2$ VCD > 100 DU, $\Omega_{SO2\_step1}$ is used as input to the lookup table to generate an updated SO$_2$ Jacobian spectrum. The iterations continue

until the results converge or the number of iterations exceeds the upper limit (15). The same data processing steps are carried out separately for the TRL, TRM, TRU, and STL SO$_2$ profiles, resulting in four final estimates of SO$_2$ VCD for each pixel.

For volcanic SO$_2$ retrievals, we start with a nominal fitting window of 313-340 nm, but drop the shortest wavelengths for pixels with large SO$_2$ loading. As demonstrated in Figure 2, saturation of large SO$_2$ absorption signals at short wavelengths leads to errors of more than 30% in the interpolated Jacobians, even for this idealized example in which

the nonlinearity in SO$_2$ absorption constitutes the only source of interpolation error. To reduce this error, we update the fitting window at each iteration step by locating the wavelength of the largest SO$_2$ Jacobian, up to 326.5 nm, and excluding all shorter wavelengths. We select 326.5 nm as the upper limit for the short end of the fitting window so that at least half of the wavelengths in the nominal window are used in the fitting. In addition, the fitting window is allowed to move to longer wavelengths only, otherwise in some cases the short end of the fitting window can change back and forth between iteration

steps, resulting in non-convergence. Our approach not only helps to minimize this particular source of interpolation error, but it also maintains sensitivity by ensuring that those wavelengths with the largest SO$_2$ Jacobians are included in the spectral fitting.

## 3 Results: comparison with the Linear Fit (LF) algorithm

In this section, we present some examples of the volcanic SO$_2$ retrievals generated with the new OMSO2VOLCANO

algorithm. The entire OMI data record has been reprocessed with this algorithm, and all data are publicly available (see Data Availability section). Here, the focus is mainly on the comparison between the OMSO2VOLCANO and the current operational LF algorithm, but we will also compare results from other algorithms wherever data are available.

### 3.1 Background regions

While much attention has been paid to explosive volcanic eruptions and their impacts on climate and aviation safety, the

importance of degassing volcanoes is now receiving increasing recognition (see section 1), despite the dearth of information on the strength of their emissions (e.g., Ge et al., 2016). OMI SO$_2$ retrievals can help to supply this information (e.g., Carn et al., 2013; McLinden et al., 2016; Fioletov et al., 2016), but ideally these retrievals need to have relatively low levels of noise and biases, to achieve sufficient sensitivity to detect degassing sources. Here we compare the global lower tropospheric (TRL) SO$_2$ VCD retrieved with the operational LF and the OMSO2VOLCANO algorithms for August 5, 2006, a day with

no major volcanic plumes or long-range transport of anthropogenic SO$_2$ in the free troposphere. As shown in Figure 3, both products detect SO$_2$ signals over large emission sources such as eastern China, the Persian Gulf, and Nyiragongo volcano in





eastern D.R. Congo, but the PCA-based OMSO2VOLCANO retrievals have much smaller bias and noise levels when compared with the LF retrievals. This is probably due to the fact that the PCA algorithm utilizes the full spectral content of OMI measurements, as opposed to just 10 wavelengths used in the LF algorithm. If we compare the standard deviations of the two retrievals over the $SO_2$-free remote Pacific, we can see that, for the majority of latitude bands from 40°S to 60°N, the

standard deviations of the OMSO2VOLCANO retrievals are ~0.2-0.3 DU, less than half of the standard deviation of LF retrievals (Figure 4). The reduced biases and noise make the new OMSO2VOLCANO dataset more sensitive and stable, providing enhanced long-term monitoring of continuously degassing sources. An example of this capability is provided in section 4.

### 3.2 Kasatochi eruption in 2008

Next we examine two major volcanic eruptions during the OMI mission. The first one, the Kasatochi eruption in August 2008, is the largest to date during the OMI era (in terms of $SO_2$ discharge by a single explosive eruption), and has been studied extensively using several UV and IR satellite instruments (e.g., Carn et al., 2016). Previous studies suggest that the eruption injected $SO_2$ to between 7 and 13 km altitude with a peak at ~10 km (Kristiansen et al., 2010; Krotkov et al., 2010; Nowlan et al., 2011; Wang et al., 2013; Yang et al., 2010; Ge et al., 2016). The estimated total $SO_2$ mass emitted by

Kasatochi varies from 1500 to 2200 kt (kiloton, $10^3$ ton), much greater than the ~850-860 kt retrieved with the LF algorithm assuming the STL (18 km) $SO_2$ profile for orbits 21635 and 21636, the two earliest OMI overpasses after the eruption (Figure 5). The estimated $SO_2$ mass for these two orbits based on LF TRM (8 km) retrievals is slightly greater, but at ~950 kt it is still biased low with respect to the other estimates by almost a factor of two, most likely due to signal saturation by high $SO_2$ loading in the young volcanic cloud.

The OMSO2VOLCANO (PCA) retrievals (Figures 5b and 5e) yield much greater estimates of $SO_2$ mass: ~1200 kt for TRU (13 km) (Figures 5c and 5f) and ~2000-2200 kt for TRM (8 km). We also note that the OMSO2VOLCANO retrieved $SO_2$ loadings for the two orbits agree to within 2% for STL and TRU and to within 10% for TRM. This suggests that the algorithm has very small cross-track biases, given that essentially the same plume was observed with different rows of the OMI instrument in these two consecutive orbits just ~1.5 hr apart. Taking the average of the TRU and TRM retrievals,

we estimate that the OMSO2VOLCANO retrievals would have given an estimated $SO_2$ mass of ~1700 kt for these two orbits, had a 10-km profile been used. This is ~10% larger than the off-line OMI ISF retrievals (1500 kt; Yang et al., 2010) and GOME-2 optimal estimation retrievals (1600 kt; Nowlan et al., 2011), but smaller than the estimate based on the $SO_2$ loss rate observed over several days following the eruption (2200 kt; Krotkov et al., 2010). The total $SO_2$ mass for OMI orbit 21636 estimated with a DOAS algorithm by Theys et al. (2015) is smaller at 1060 kt, but a plume height of 15 km was

assumed in that study.

For OMI orbit 21650 that captured the Kasatochi plume about 24 hr later, the total $SO_2$ mass estimated with OMSO2VOLCANO is greater than that for orbit 21636 (~1300 kt vs. ~890 kt for STL, ~1500 kt vs. ~1200 kt for TRU, ~2300 kt vs. ~2000 kt for TRM). This apparent increase in the $SO_2$ loading over time has also been reported for other OMI





retrievals for Kasatochi including the LF (1300 kt vs. 850 kt on the first day for STL retrievals), ISF (1600 kt vs. 1500 kt on the first day), and DOAS (1220 kt vs. 1060 kt on the first day) algorithms, and previously for other eruptions measured by different instruments (e.g., TOMS retrievals for the Pinatubo eruption), suggesting that the $SO_2$ mass may still be underestimated for highly concentrated plumes observed shortly after eruptions. This is perhaps due to the combined effects

of $SO_2$ signal saturation and high volcanic ash and aerosol loadings. Increases in measured volcanic $SO_2$ loading beyond the first day of observation have also been attributed to conversion of emitted $H_2S$ to $SO_2$ (e.g., Rose et al., 2000), and the new OMSO2VOLCANO retrievals promise more accurate assessments of this possibility by better eliminating the uncertainties due to $SO_2$ signal saturation in fresh volcanic clouds. In the Kasatochi case, a preliminary mass balance suggests that not all the observed $SO_2$ increase (>100 kt) can be accounted for by conversion of detected $H_2S$ (29±10 kt of $H_2S$ reported in the

Kasatochi plume by Clarisse et al. (2011) would yield a maximum of ~74 kt of $SO_2$), but there are also issues with satellite sensitivity to $H_2S$ (see Clarisse et al., 2011).

Regarding the maximum $SO_2$ VCDs, for orbit 21635, the highest OMSO2VOLCANO VCDs are 238, 388, and 685 DU for STL, TRU, and TRM retrievals, respectively. In comparison, the maximum LF VCDs are smaller at 220 and 379 DU for STL and TRM, respectively. Similarly for orbit 21636, the OMSO2VOLCANO peak VCDs are also greater at 260, 350,

and 644 DU for STL, TRU, and TRM, respectively, compared with 246 and 274 DU for STL and TRM retrievals in the LF product. The DOAS algorithm developed by Theys et al. (2015) appears to produce greater peak $\Omega_{SO2}$ for this case, with $\Omega_{SO2}$ maxima of 382 DU for OMI orbit 21635, and 565 DU for GOME-2 data from the same day, despite assuming a 15-km $SO_2$ profile and giving a smaller estimate of the overall $SO_2$ mass for the plume (see above). The DOAS algorithm uses the spectral window of 312-326 nm for baseline slant column $SO_2$ amounts (SCD1). It also retrieves additional SCDs using

alternative windows (325-335 nm for SCD2 and 360-390 nm for SCD3) and determines the final fitting window based on the comparison between the SCDs. For most pixels, the baseline window is retained as the final window, but for pixels with moderate (SCD1 > 40 DU and SCD2 > SCD1) and large (SCD2 > 250 DU and SCD3 > SCD2) $SO_2$ signals, the longer wavelength windows of 325-335 nm and 360-390 nm are used, respectively (*cf.* Theys et al., 2015 for details). The $SO_2$ cross sections in 360-390 nm are over an order of magnitude smaller than those at the wavelengths used in PCA retrievals. Such

large differences in the cross sections may contribute to the different peak $SO_2$ VCDs between the PCA and DOAS retrievals, but a more detailed comparison between the two algorithms will be necessary to fully understand their differences.

**3.3 Sierra Negra eruption in 2005**

The second example we examined was the effusive eruption of Sierra Negra (Galápagos Islands) in 2005. This eruption began on October 22 of that year, and injected large amounts of $SO_2$ into the lower troposphere with little ash (Yang et al.,

2009), thus representing different conditions to the explosive Kasatochi eruption. The Sierra Negra volcanic plume was measured using OMI LF TRL retrievals for orbit 6779 on October 23, yielding a maximum $SO_2$ VCD of 246 DU and a total $SO_2$ mass of 363 kt (Figure 6a). The OMSO2VOLCANO TRL retrievals (Figure 6b) for the same orbit reveal a similar spatial distribution of the $SO_2$ plume, but give a much greater total $SO_2$ mass of 694 kt, almost double that of the LF



algorithm. The maximum $SO_2$ VCD retrieved with the OMSO2VOLCANO algorithm is extraordinarily high at ~1125 DU, the largest detected to date by satellites. A similarly large VCD has also been retrieved for the same case with the off-line OMI ISF algorithm (>1100 DU; Yang et al., 2009). The good agreement between the OMSO2VOLCANO and ISF algorithms suggests that both can produce less-biased retrievals than the LF algorithm, but unlike the ISF algorithm,

OMSO2VOLCANO has been implemented operationally and doesn't require online radiative transfer calculations and instrument-specific corrections to the radiance data (i.e., soft calibration).

**4 Data continuity with the Suomi-NPP OMPS instrument**

Now already in its 12[th] year of service, OMI is expected to continue to provide global monitoring of volcanic $SO_2$. However, the quality and spatial coverage of OMI measurements are presently limited by instrument issues that have developed over

time. In particular, since 2009 about half of the 60 OMI rows have been rendered unusable by a partial blockage of the field of view (i.e., the OMI row anomaly; http://projects.knmi.nl/omi/research/product/rowanomaly-background.php). Several satellite instruments currently in orbit or to be launched in the near future will continue the UV satellite volcanic $SO_2$ data record (Carn, 2015; Carn et al., 2016). Particularly suitable for this task is the OMPS nadir mapper (OMPS-NM) on board the Suomi-NPP spacecraft (Flynn et al., 2014; Seftor et al., 2014). This platform was launched in 2011 and is flying in a sun-

synchronous orbit with a local afternoon overpass time close to that of the Aura satellite (OMI). OMPS has already made several years' worth of measurements that overlap the OMI mission. This makes it possible to conduct extensive comparisons between retrievals from the two instruments, in order to evaluate their consistency. Like OMI, the 2-D "push-broom" CCD detector of OMPS-NM covers the entire globe on a daily basis, allowing the total $SO_2$ mass to be estimated for complete volcanic clouds. However, there are challenges in the construction of a consistent volcanic $SO_2$ dataset between the

two instruments, given the coarser spatial ($50 \times 50$ km$^2$ vs. $13 \times 24$ km$^2$ at nadir) and spectral (1 nm vs. ~0.5 nm) resolution offered by OMPS-NM. Furthermore, differences in instrument calibration also need to be taken into account to minimize inter-instrument biases.

Our PCA-based retrieval approach can help to overcome these challenges, as it has small biases and requires no explicit instrument-specific corrections to the radiance data. Indeed, in a separate study (Zhang et al., 2016), we showed that

the PCA PBL $SO_2$ algorithm was able to produce nearly unbiased retrievals for regional $SO_2$ pollution events between OMI and OMPS. Here, we examine the feasibility of using OMPS to continue the volcanic $SO_2$ emission dataset produced from OMI, for both continuously degassing volcanoes and large eruptions. The OMPS PCA volcanic $SO_2$ algorithm used here is almost identical to the OMSO2VOLCANO algorithm, with the exception of two implementation details: 1) we used fewer PCs in the spectral fitting (up to 15 PCs for OMPS, compared with 20 for OMI), since OMPS has approximately half the

wavelengths of OMI in the same spectral fitting window; 2) we extended the OMPS retrievals to pixels with solar zenith angles up to 75° to gain more spatial coverage near the edge of the swath at high latitudes in winter months.





### 4.1 Long-term monitoring of SO₂ degassing from Kīlauea volcano

Kīlauea volcano is the most active of the five volcanoes on the island of Hawaiʻi and has been in a state of near-continuous effusive eruption since 1983. Lower tropospheric $SO_2$ plumes emitted from Kīlauea have been detected routinely by OMI since the start of the Aura mission (e.g., Carn et al., 2013). Here, we compare the PCA volcanic $SO_2$ retrievals over the

Hawaiʻi region from both OMI and OMPS instruments (Figure 7). To calculate the daily $SO_2$ mass within the domain around Kīlauea (16-21°N, 154-160°W), we first gridded OMI and OMPS TRL $SO_2$ retrievals to the same 0.5°×0.5° horizontal resolution and then calculated the total $SO_2$ loading for each instrument by summing the $SO_2$ mass in all grid cells with $SO_2$ VCD > 0.4 DU. This threshold was selected to include only signals that are at least twice the retrieval noise level for the area (~0.2 DU, see Figure 4).  To ensure consistency in sampling, only the first 24 rows of OMI (considered to be unaffected by

the row anomaly) were used, and only those days with at least 90% spatial coverage by these 24 rows were included in the time series plot in Figure 7a. We applied the same $SO_2$ threshold and spatial coverage constraints to the OMPS data, but used all 36 rows of the OMPS-NM sensor. As shown in Figure 7a, OMI has detected substantial variability in the $SO_2$ loading over the area near Kīlauea, with a calm period until March 2008 and an active period during 2008-2010, followed by another relatively calm period. This is qualitatively consistent with the estimates by Carn et al. (2013) and Ge et al. (2016) using the

OMI LF retrievals, and also with ground observations (Elias and Sutton, 2012). OMPS provided more frequent measurements during 2012-2015 due to its more complete spatial coverage. The OMPS $SO_2$ time series compares well with the OMI record for the period. A regression analysis (Figure 7b) confirms that OMPS estimates of daily regional $SO_2$ mass are well correlated with OMI estimates, with a correlation coefficient ($r$) of 0.89 and a slope of 1.12, indicating slight overestimates by OMPS as compared with OMI. Analysis using reduced major-axis regression yields a slope of 1.25, still

suggesting overall consistency between OMI and OMPS. We also investigated the correlation between the spatial distributions of the OMI and OMPS PCA retrievals on a daily basis (Figure 7c) and found that for the ~200 days when both instruments detected at least 0.5 kt of $SO_2$ over the area, 84% of them had $r$ of at least 0.5 (66% had $r$ > 0.7).  Only five of these days had $r$ < 0.3 (02/05/2012, 10/02/2012, 05/14/2013, 11/06/2013, and 11/09/2014) and on all five days, the core of the plume was covered by OMI pixels near the nadir position, but by OMPS pixels near the edge of the swath. This

comparison demonstrates that OMPS is capable of extending the long-term OMI record of $SO_2$ emissions from degassing volcanoes.

### 4.2 Kelut Eruption in 2014

The violent explosive eruption of Mt. Kelut in Indonesia on the night of February 13, 2014 injected a volcanic plume to altitudes of ~16-17 km and above (Kristiansen et al., 2015). OMSO2VOLCANO $SO_2$ retrievals for the OMI overpass on the

following day (Figure 8a) were able to capture the plume, but the estimated total $SO_2$ mass of ~131 kt is certainly biased low, as a sizeable part of the plume was obscured by the OMI row anomaly. The OMPS PCA retrievals (Figure 8b) produce a very similar spatial pattern to the OMI retrievals, and the complete plume coverage by OMPS yields a greater estimate of





total SO$_2$ (179 kt). On the other hand, the OMPS retrievals lack some of the spatial detail offered by OMI and have a smaller maximum SO$_2$ VCD (47.5 DU vs. 69.3 DU); this is probably mainly due to its coarser spatial resolution. Given that OMI and OMPS overpasses were only 30 min apart in this case, we can combine the OMI and OMPS retrievals to produce an integrated SO$_2$ map that covers the plume in its entirety whilst providing fine spatial resolution where OMI data are available

(Fig. 8c). This is facilitated by the very good agreement between OMI and OMPS PCA retrievals, and involves assigning the SO$_2$ VCD from the closest OMPS pixel (based on pixel center coordinates) to each OMI row anomaly pixel. In fact, the total SO$_2$ mass estimated from the combined OMI+OMPS map (174 kt; Figure 8c) agrees with the OMPS-only estimate to within 3%. The Kelut example demonstrates that our OMPS PCA data can supplement OMI retrievals to provide complete and consistent coverage for large volcanic eruptions.

**5 Conclusions**

In summary, we have developed a new generation OMI volcanic total SO$_2$ column amount dataset using an algorithm based on a principal component analysis (PCA) technique. The new algorithm, OMSO2VOLCANO, directly extracts spectral features (in the form of principal components, PCs) from OMI radiances that are associated with various geophysical processes (e.g., O$_3$ absorption, RRS) and measurement details (e.g., wavelength shift). The algorithm then fits these PCs and

SO$_2$ Jacobians (representing instrument sensitivity to a perturbation in column SO$_2$) to the measured radiances in order to estimate SO$_2$ while minimizing the impacts of various interferences on SO$_2$ retrievals. A table-lookup approach was developed to determine SO$_2$ Jacobians under various conditions such as measurement geometry (solar zenith angle, viewing zenith angle, and relative azimuth angle), SO$_2$ amount, O$_3$ amount (from OMI or OMPS total O$_3$ products), and the reflectivity of the underlying surface. To first order, the effects of clouds and aerosols are accounted for using a simple

Lambertian equivalent reflectivity (SLER) derived at longer wavelengths (342.5, 354.1, 367.04 nm) and then extrapolated to our nominal fitting window of 313-340 nm. For very large volcanic plumes, SO$_2$ absorption may saturate at shorter wavelengths, leading to large interpolation error. To circumvent this problem, the spectral fitting window is updated dynamically at each iteration step by locating the wavelength with the maximum SO$_2$ Jacobian in the nominal window and dropping all shorter wavelengths. In the absence of information on SO$_2$ plume height, retrievals of SO$_2$ total column amounts

are given for four different pre-defined SO$_2$ profiles with peaks at 3 km (TRL), 8 km (TRM), 13 km (TRU), and 18 km (STL), representing typical altitudes of plumes from non-eruptive volcanic degassing, moderate eruptions, and explosive eruptions (TRU and STL), respectively.

Comparisons with the current operational OMI volcanic SO$_2$ product based on the linear fit (LF) algorithm and other satellite data sets suggest that the new OMSO2VOLCANO dataset has significantly lower biases and noise with

respect to the LF, with the background noise reduced by half or more over the remote Pacific. For the Kasatochi eruption in 2008, the OMSO2VOLCANO retrievals give an estimated SO$_2$ total mass of ~1700 kt from OMI observations shortly after the eruption, a factor of two greater than LF estimates and generally in agreement with OMI iterative spectral fitting (ISF,





Yang et al., 2010) and DOAS (Theys et al., 2015) retrievals as well as GOME-2 optimal estimation retrievals (Nowlan et al., 2011). For the Sierra Negra eruption in 2005, the OMSO2VOLCANO algorithm detects a peak $SO_2$ column amount of over 1100 DU. This value is consistent with that obtained using the off-line ISF algorithm (Yang et al. 2009), and this concurrence suggests that for this case, the LF algorithm underestimates peak $SO_2$ amounts by a large margin. Overall, the

new OMSO2VOLCANO dataset has much improved data quality for both quiescent degassing volcanoes and large eruptions, when compared with the LF product. Unlike the ISF algorithm, the OMSO2VOLCANO algorithm has been implemented operationally as it requires far less computational resources and also does not rely on instrument-specific soft calibration.

To extend the 10-year OMI volcanic $SO_2$ data record and to mitigate the loss of OMI spatial coverage due to the

row anomaly, we have implemented the same PCA volcanic $SO_2$ algorithm with the Suomi-NPP OMPS instrument nadir mapper. Despite its coarser spatial and spectral resolution, the PCA retrievals with OMPS are highly consistent with those from the OMSO2VOLCANO dataset. For the area around the continuously degassing Kīlauea volcano in Hawai'i, the OMPS and OMI estimates of daily total $SO_2$ loading are highly correlated ($r = 0.89$, slope = 1.12). For large volcanic eruptions, OMPS and OMI PCA retrievals can be merged to achieve complete spatial coverage and fine measurement

details, as demonstrated for the Kelut eruption in 2014. The estimated total $SO_2$ mass within the Kelut plume from the combined OMI-OMPS data agrees with the OMPS-only data to within 3%, again indicating very good agreement between OMI and OMPS retrievals.

Overall, the new PCA-based OMSO2VOLCANO dataset offers enhanced sensitivity to small volcanic degassing sources and more accurate retrievals for large volcanic plumes as compared with the current operational OMI volcanic $SO_2$

product. The new dataset will help to improve our understanding of the impacts of volcanic emissions, and the new record will be continued with existing instruments and extended with future sensors such as the next generation of OMPS on the Joint Polar Satellite System (JPSS) spacecraft and TROPOMI (the TROPospheric Monitoring Instrument) on the Copernicus Sentinel 5 Precursor.

**Data availability**

The new-generation OMI volcanic (TRL, TRM, and STL) $SO_2$ product based on the OMSO2VOLCANO algorithm (OMSO2 v1.3.0) is publicly available from the NASA Goddard Earth Sciences (GES) Data and Information Services Center (DISC) (http://disc.sci.gsfc.nasa.gov/Aura/data-holdings/OMI/omso2_v003.shtml). The OMI TRU $SO_2$ data are upon request from the corresponding author.

**Acknowledgements**

The authors acknowledge the NASA Earth Science Division (ESD) Aura Science Team program (managed by Ken





Jucks) for funding of OMI SO$_2$ product development and analysis. The Dutch and Finnish built OMI instrument is part of the NASA's Earth Observing System (EOS) Aura satellite payload. The OMI project is managed by the Royal Meteorological Institute of the Netherlands (KNMI) and the Netherlands Space Agency (NSO). CL acknowledges partial support from NASA's Earth Science New Investigator Program in developing the OMPS SO$_2$ algorithm (Grant # NNX14AI02G).

The authors would also like to thank the NASA OMPS ozone Product Evaluation and Test Element (PEATE) team for updating the OMPS calibration and producing the OMPS Level 1b and Level 2 O$_3$ data used in this analysis. We thank the OMI calibration team, led by KNMI, for the calibrated OMI Level 1b data used here.

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

Zhang, Y., Li., C., Krotkov, N. A., and Joiner, J.: Continuation of long-term global $SO_2$ pollution monitoring from OMI to OMPS, Atmos. Meas. Tech., to be submitted, 2016.

**Tables**

Table 1. Nodes of the solar zenith angle (SZA), viewing zenith angle (VZA), and $SO_2$ column amount,

as used in the pre-computed $SO_2$ Jacobians lookup tables.

| Parameter | Nodes | | | | | | | | | | | | | |
|---|---|---|---|---|---|---|---|---|---|---|---|---|---|---|
| SZA | 0° | | 15° | | 30° | | 45° | | 60° | | 70° | 77° | | 81° |
| VZA | 0° | | 15° | | 30° | | 45° | | 60° | | 70° | 75° | | 80° |
| $SO_2$ (DU) | 0 | 1 | 5 | 10 | 50 | 100 | 200 | 300 | 400 | 500 | 600 | 700 | 800 | 900 | 1000 |



# Figures

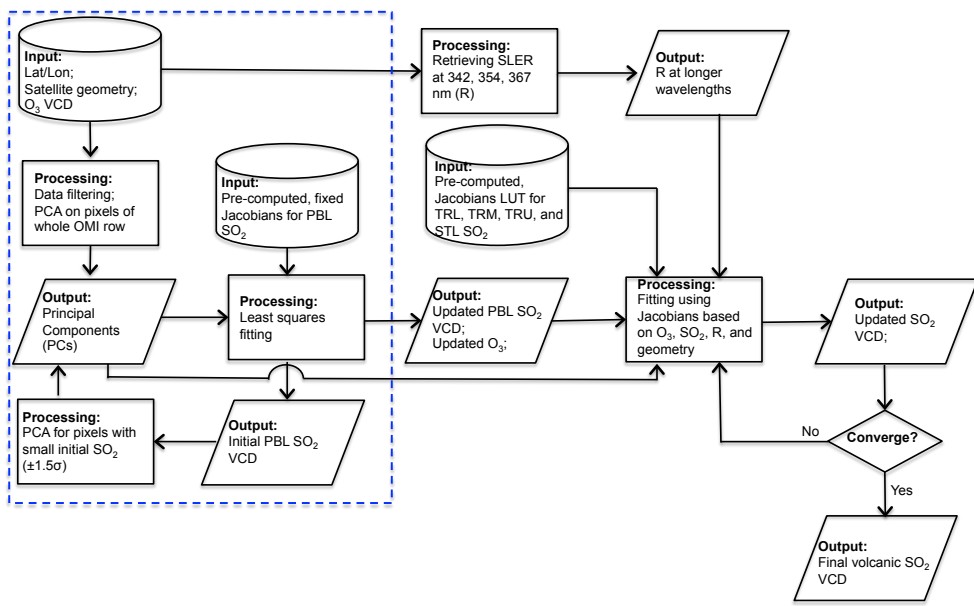

**Figure 1: A flow chart of the new OMI volcanic SO₂ PCA retrieval algorithm (OMSO2VOLCANO). The first part of the algorithm, enclosed in the dashed blue box, is essentially identical to the operational OMI planetary boundary layer (PBL) SO₂**
5 **algorithm (Li et al., 2013), and provides input to the second part of algorithm that performs iterative spectral fitting to retrieve volcanic SO₂ VCD.**





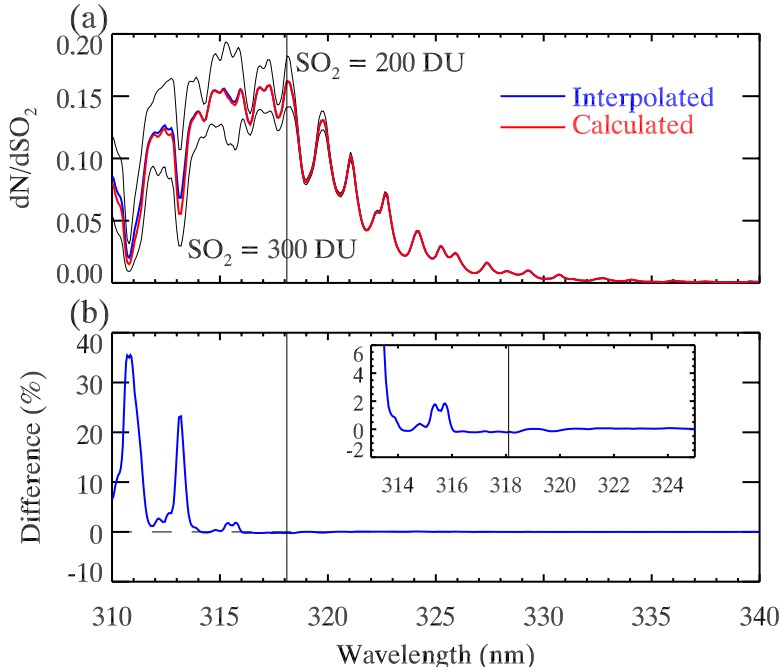

**Figure 2: (a) A comparison of SO₂ Jacobians for an idealized pixel (SZA = 30°, VZA = 45°, RAZ = 90°, R = 0.05, an SO₂ plume centered at 18 km with Ω_SO2 = 250 DU, and a mid-latitude O₃ profile with Ω_O3 = 375 DU) derived from direct calculation using VLIDORT (red line) and interpolation (blue line) from VLIDORT calculations for two bracketing SO₂ nodes (200 DU and 300**

5   **DU) shows sizable (b) interpolation errors at wavelengths < 315 nm, caused by nonlinearity owing to the saturation of SO₂ absorption signals. The relative difference between interpolated and directly calculated SO₂ Jacobians exceeds 30% at wavelengths < 315 nm and remains substantial at about 2% at ~316 nm (see insert). The wavelength with the maximum interpolated SO₂ Jacobian, or the start of the fitting window for this particular example, is ~318 nm, and is marked with a vertical solid line in both (a) and (b).**





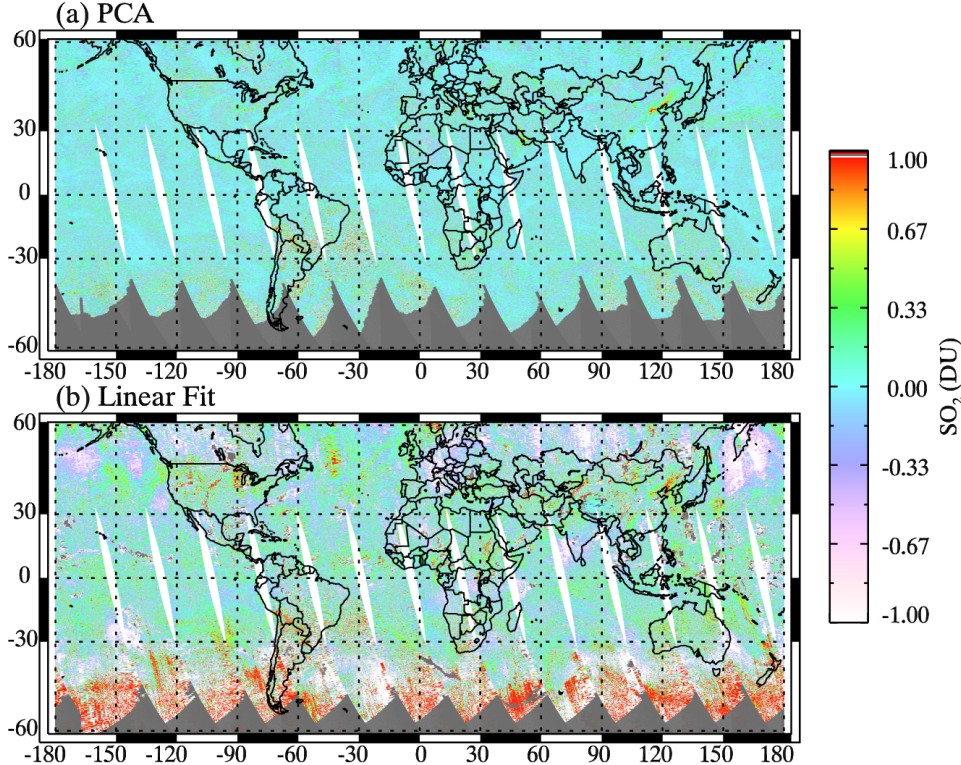

**Figure 3:** (a) Level 2 OMI SO$_2$ total vertical column density for August 5, 2006, retrieved with the OMSO2VOLCANO (PCA) algorithm assuming a lower tropospheric SO$_2$ plume with center of mass at 3 km altitude (TRL profile). (b) Same as (a) but for the current operational linear fit (LF) algorithm. Only pixels within the center 56 rows are plotted for both products.





**Figure 4: Standard deviation of TRL total SO$_2$ vertical column density retrieved for 10° latitude bands over the remote Pacific (170-180°W) on August 5, 2006 with the linear fit (blue line) and OMSO2VOLCANO (PCA, red line) algorithms.**





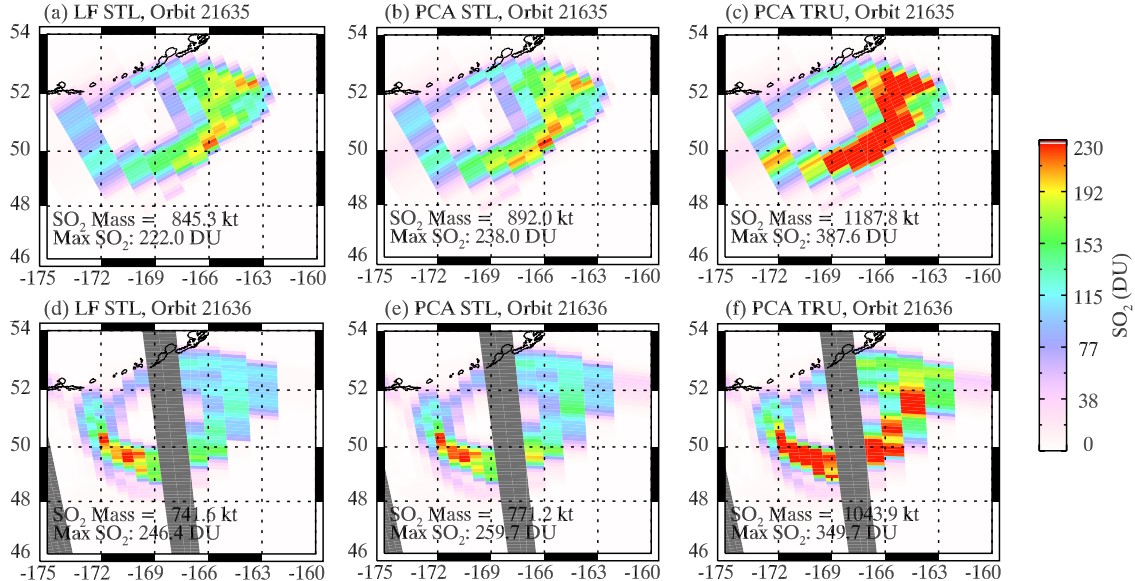

**Figure 5: (a) OMI linear fit SO$_2$ retrievals assuming a lower stratospheric (STL) SO$_2$ profile for orbit 21635, the first OMI overpass after the Kasatochi eruption on August 8, 2008. (b-c) Same as (a) but for OMSO2VOLCANO (PCA) retrievals assuming (b) the STL SO$_2$ profile and (c) an upper tropospheric (TRU) SO$_2$ profile. (d) Linear fit STL retrievals for orbit 21636, the second OMI overpass after the eruption. (e-f) Same as (d) but for OMSO2VOLCANO (e) STL and (f) TRU retrievals. The gray-shaded rows in (d-f) are masked due to the OMI row anomaly. If retrievals for these masked rows are included in the calculation of the total loading, the total LF STL SO$_2$ mass in the domain for orbit 21636 is 864.7 kt (*cf.* Krotkov et al., 2010), whereas PCA total mass in the domain is 897.2 kt and 1204.9 kt for STL and TRU retrievals, respectively.**





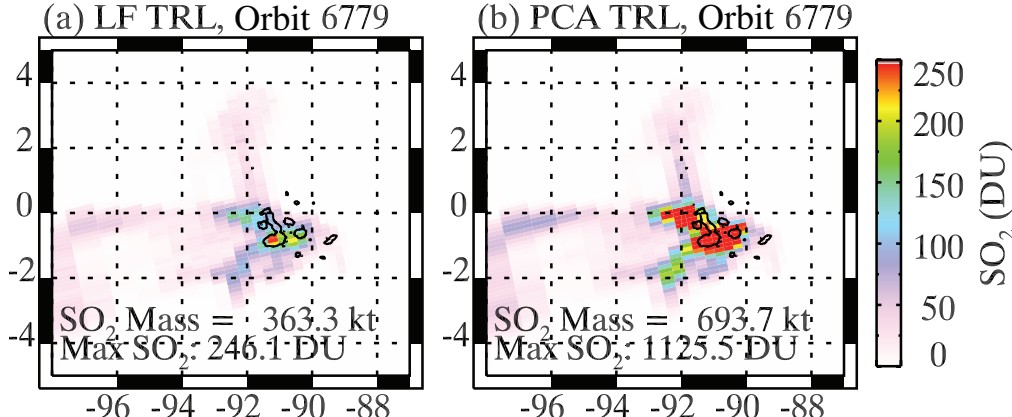

**Figure 6: (a) OMI LF SO₂ retrievals assuming a lower tropospheric (TRL) SO₂ profile over the Galápagos Islands, from orbit 6779 on 23 October 2005, after the eruption of Sierra Negra volcano on the previous day. (b) Same as (a) but for OMSO2VOLCANO (PCA) TRL retrievals.**





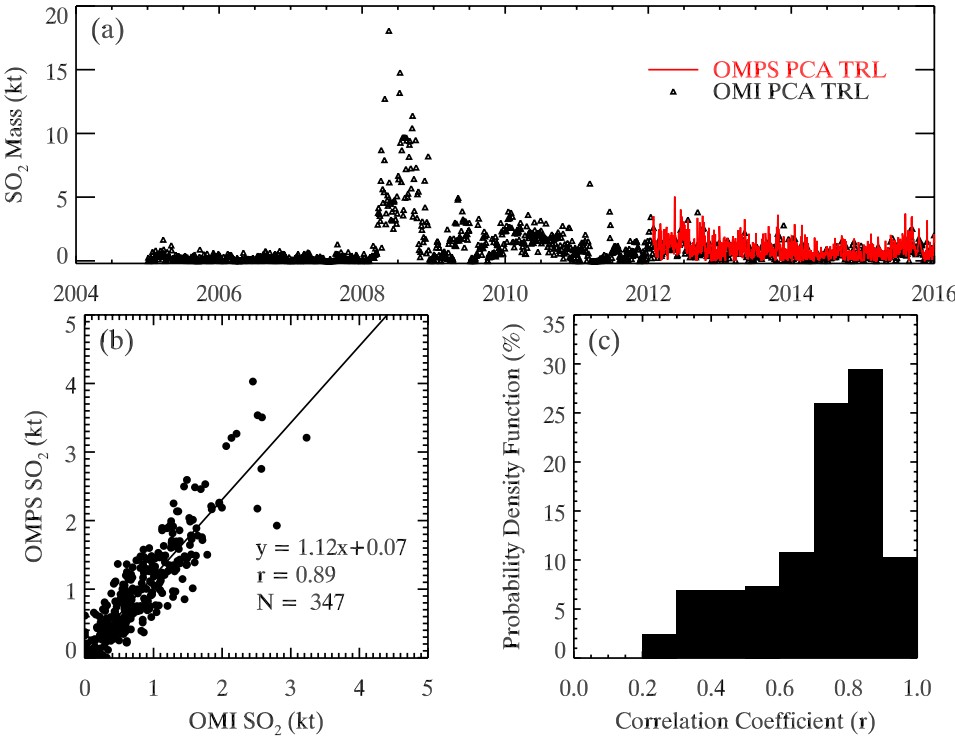

**Figure 7: (a) Daily total SO₂ mass in a domain (16-21°N, 154-160°W) around the Kīlauea volcano in Hawai'i, retrieved with OMI and OMPS using the PCA algorithm assuming a lower tropospheric (TRL) SO₂ profile. Only days with at least 90% coverage of the domain by either OMI (the first 24 rows) or OMPS (all 36 rows) are shown. (b) Scatter plot between OMI- and OMPS-estimated total SO₂ over the domain for days with at least 90% spatial coverage by both OMI (the first 24 rows) and OMPS. (c) The probability density function of the daily spatial correlation coefficient between OMI and OMPS TRL SO₂, both gridded at 0.5°×0.5° resolution. Only those days (~200 in all) when both instruments covered over 90% of the domain and detected over 0.5 kt of SO₂ are included.**



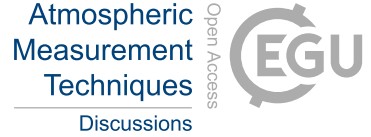

(a)

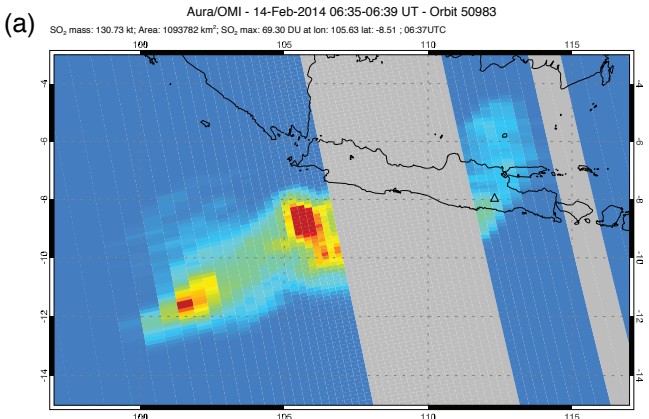

(b)

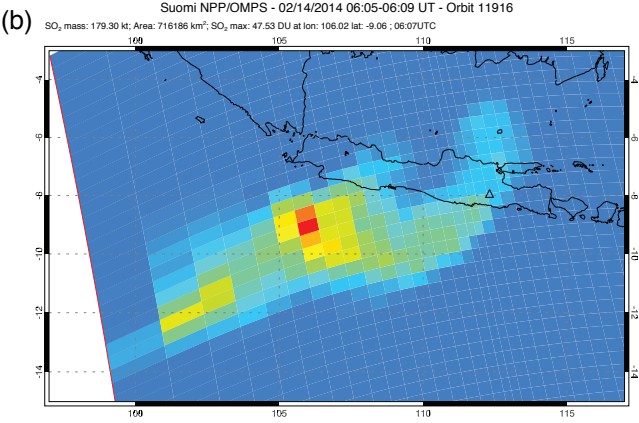

(c)

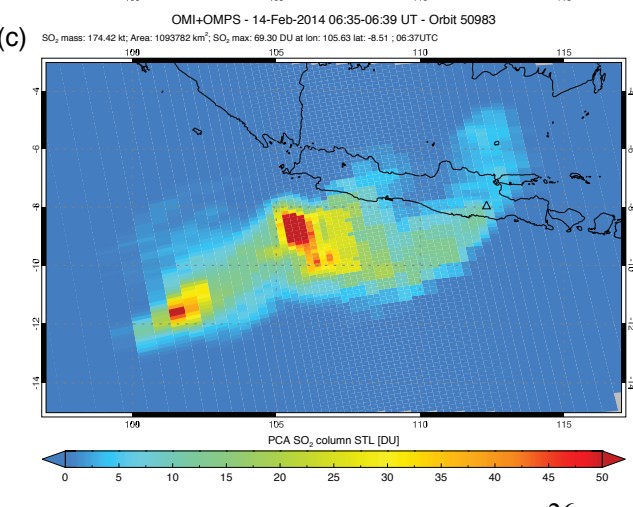





**Figure 8: (a) The volcanic SO$_2$ plume from the Kelut eruption on February 13, 2014 retrieved using the OMI PCA algorithm (OMSO2VOLCANO) assuming the STL SO$_2$ profile on the following day. Shaded areas are masked due to the OMI row anomaly. (b) Same as (a) but for OMPS PCA retrievals. (c) Same as (a) but for the merged OMI and OMPS PCA STL SO$_2$ retrievals.**