# Peer review of "New-generation NASA Aura Ozone Monitoring Instrument (OMI) volcanic SO₂ dataset: Algorithm description, initial results, and continuation with the Suomi-NPP Ozone Mapping and Profiler Suite (OMPS)"

_Atmospheric Measurement Techniques, 2016_

## Referee Comment (RC1) · Anonymous Referee #2 · 21 Nov 2016

Title: New-generation NASA Aura Ozone Monitoring Instrument (OMI) volcanic SO2 dataset: Algorithm description, initial results, and continuation with the Suomi-NPP Ozone Mapping and Profiler Suite (OMPS).

Li et al. reported on a new algorithm to retrieve volcanic SO2 from OMI and OMPS. It will have a great value for users of OMI operational product. I recommend publication after addressing the following comments:

[Figure]

1) Page 3, last line: Is it true that PCA technique allows for studying regional trends in SO2? I would think that even BRD SO2 product would be able to track those trends. Please rephrase.

2) Page 5, l1-2: what is 'significant correlation'? It would be good to give more details. Generally speaking , it is not very clear what is the correlation between SO2 and all other PCs and why it is apparently not introducing biases in the data (also for anthropogenic SO2). An illustration of the correlation coefficients (between the PCs and SO2) and discussion is needed.

3) Equation 2: Please specify what are the Intensity terms (I0, I1, I2), it is not mentioned. Also an analytic dependence with relative azimuth angle is given without any explanation on why it is so. It is not fully clear why the formulation with the different terms is used. I presume it is to reduce the size of the lookup table for the relative azimuth angle (please clarify). It is also not clear why the fitting of R is not performed directly on total intensity (the left term of eq.2). This could be done using a small LUT (because of neglect of gaseous absorption).

4) P6, l 22: it is written "..extrapolate R to shorter wavelengths" while in Fig 1 it reads "longer wavelengths"

5) P6, l24-33: why is a multiple dimension linear interpolation not done (in one step)?

6) P6, l10: the interpolation error could be reduced by having finer LUT grids and / or use higher order interpolation (e.g. spline) without increasing unreasonably the LUT size. The author should justify why this was not preferred. When changing fitting window, in principle a new calculation of PCs should be performed. Is it the case or does the algorithm simply sample the original PCs over the new fitting range? If so, what is the impact of this simplified approach on the results?

7) Ge et al., 2016 cannot be found in the references list.

8) Section 3.1 contains no new information compared to previous work (Li et al., 2013).

The author should consider replacing this section by other results.

9) In section 3.2, it would be interesting to show the inverted reflectivity R and illustrate the wavelength range used (smallest wavelength).

10) Section 3.2 and 3.3: it is not clear why in one case (Kasatochi) the LF and PCA STL results are so close (is PCA really improving the situation?) and for the other case (Sierra Negra) the two algorithms produce results that are so different. Please expend the discussion.

11) Section 3.2, Fig 5: it is surprising to see a strong change between PCA TRU and PCA STL. The expected change in measurement sensitivity from 13 to 18 km should be small and is incompatible with this observed change.

12) P10, l19: the author should describe better what are those challenges.

13) P12, l2: The author states that differences between OMI and OMPS SO2 VCDs are attributed to differences in spatial resolution but, based on Fig 8, it is hard to believe. It is clear that there are patterns (with higher SO2) in the SO2 map which are typically of the same size or larger than the OMPS pixels. Please clarify.

Typos

-page 2, l20: "due the small .." –> "due to the small.."

-page 5, l7 and 19: the same notation (I) is used for two different quantities (according to the text): sun-normalized earthshine radiance and backscatter radiances at TOA. Please clarify.

---

## Referee Comment (RC2) · Anonymous Referee #1 · 29 Nov 2016

Review of "New-generation NASA Aura Ozone Monitoring Instrument (OMI) volcanic SO2 dataset: Algorithm description, initial results, and continuation with the Suomi-NPP Ozone Mapping and Profiler Suite (OMPS) " by Li et al.

Li et al. present a variant on their very successful PCA boundary-layer SO2 retrieval algorithm, aimed here at retrieving volcanic SO2, and apply it to OMI and OMPS. For larger SO2 loading they utilize long wavelengths in their retrieval. The authors find greatly reduced retrieval noise, and removal of a high bias, with this product. Its

successful application to OMPS will help ensure a continuation of the OMI volcanic SO2 data record. This is clear and well written and represents an advance in the retrieval of SO2 from UV/vis satellite spectra. I recommend it be published once the reviewers address the points given below:

Page 12, line 24: "In the absence of information on SO2 plume height . . ." - OMI should have information on plume height in its spectra, at least for larger eruptions. What about retrieving SO2 plume height? This was demonstrated previously for OMI by Yang et al. (2009). Presumably this knowledge would greatly reduce one of the larger sources of error for users. Please address this.

Yang, K., X. Liu, N. A. Krotkov, A. J. Krueger, and S. A. Carn (2009), Estimating the altitude of volcanic sulfur dioxide plumes from space borne hyper-spectral UV measurements, Geophys. Res. Lett., 36, L10803, doi:10.1029/2009GL038025

Section 3: More detailed/quantitative/spatial comparisons should be made with GOME-2. E.g., Figure 5 and figure 7. GOME2 is mentioned in passing but real comparisons would provide additional confidence in this new product (different sensor + different algorithm). Provide GOME2 VCD maps for one of the eruptions studied.

Page 5, line 11: change "computationally too expensive" to "too computationally expensive"

---

## Author Comment (AC1) · 24 Dec 2016

Response to review of "*New-generation NASA Aura Ozone Monitoring Instrument (OMI) volcanic SO2 dataset: Algorithm description, initial results, and continuation with the Suomi- NPP Ozone Mapping and Profiler Suite (OMPS)*" (doi:10.5194/amt-2016-221).

Referees' comments in *Italic*, Responses in blue

**Anonymous Referee #2**

*Title: New-generation NASA Aura Ozone Monitoring Instrument (OMI) volcanic SO2 dataset: Algorithm description, initial results, and continuation with the Suomi-NPP Ozone Mapping and Profiler Suite (OMPS).*

*Li et al. reported on a new algorithm to retrieve volcanic SO2 from OMI and OMPS. It will have a great value for users of OMI operational product. I recommend publication after addressing the following comments:*

We thank the referee for the review and for raising several good points. We have made changes to the manuscript. Please see below our responses to the specific comments.

*1) Page 3, last line: Is it true that PCA technique allows for studying regional trends in SO2? I would think that even BRD SO2 product would be able to track those trends. Please rephrase.*

Thank you for pointing this out. This point mainly reflects a recent study by He et al. (2016). They found that, without conducting extensive bias correction, it was much easier to derive the regional trend over the eastern U.S. using the PCA data than using the BRD data. We have added the reference and discussion in the revised manuscript.

*2) Page 5, l1-2: what is 'significant correlation'? It would be good to give more details. Generally speaking , it is not very clear what is the correlation between SO2 and all other PCs and why it is apparently not introducing biases in the data (also for anthropogenic SO2). An illustration of the correlation coefficients (between the PCs and SO2) and discussion is needed.*

We have included a figure in the supplemental information showing the correlation coefficients between the PCs and $SO_2$ for a row affected by the Kasatochi plume and another row outside of the plume. As shown in the figure, one particular PC is correlated at the 95% confidence level and apparently has $SO_2$ signatures that should be excluded in the fitting. Other PCs have much smaller correlation coefficients and we don't expect them to cause over-fitting or biases in the retrieved $SO_2$. In our previous paper (Li et al., 2013), we also noted that the first 3 PCs were associated with known geophysical processes (such as $O_3$) and they are always included in the fitting. We have added the discussion in the revised manuscript.

*3) Equation 2: Please specify what are the Intensity terms (I0, I1, I2), it is not mentioned. Also an analytic dependence with relative azimuth angle is given without any*

*explanation on why it is so. It is not fully clear why the formulation with the different terms is used. I presume it is to reduce the size of the lookup table for the relative azimuth angle (please clarify). It is also not clear why the fitting of R is not performed directly on total intensity (the left term of eq.2). This could be done using a small LUT (because of neglect of gaseous absorption).*

The intensities I0, I1, I2 are Fourier expansion coefficients in azimuthal angle that in the case of Rayleigh scattering have only 2 terms. Equation 2 allows for the de-coupling of the multiple scattering and surface reflection for the case of Rayleigh atmosphere bounded by Lambertian surface (Dave, 1964). In this case, it allows for the radiative transfer equation (RTE) to be solved efficiently and accurately, with just 2 azimuthal harmonics required.

Equation 2 has been used widely in satellite backscattered UV (BUV) retrievals of ozone and other minor gaseous components (e.g., Bhartia and Wellemeyer, 2002), where multiple Rayleigh scattering plays a major role in atmospheric radiative transfer. Indeed, it allows us to reduce the dimension of the LUTs used in operational retrievals to exclude azimuth dimension. This is especially important for spectral fitting techniques where spectral Jacobians calculation demands much of the computation, as is the case for our PCA retrieval algorithm. We have added some of the above discussion to the revised manuscript.

As for the R, we use equation 2 to calculate the radiances from the LUT and match the results with the measured radiances. We do neglect gas absorption and use a smaller LUT (by using a fixed $O_3$ profile for the LUT) and we have clarified this in the revised manuscript.

Bhartia, P. K., and Wellemeyer, C. W.: OMI TOMS-V8 Total O3 Algorithm, Algorithm Theoretical Baseline Document: OMI Ozone Products, edited by P. K. Bhartia, vol. II, ATBD-OMI-02, version 2.0, 2002, available at http://eospso.nasa.gov/sites/default/files/atbd/ATBD-OMI-02.pdf
Dave, J. V.: Meaning of successive iteration of the auxiliary equation of radiative transfer, Astrophys. J., 140, 1292-1303, 1964.

*4) P6, l 22: it is written "..extrapolate R to shorter wavelengths" while in Fig 1 it reads "longer wavelengths" .*

We have updated Fig. 1 to make it more consistent.

*5) P6, l24-33: why is a multiple dimension linear interpolation not done (in one step)?*

The interpolation needs to be done for several hundred wavelengths across the spectral range. By doing linear interpolation in multiple steps and only reading in the necessary bracketing nodes, the memory usage is kept at a relatively low level. In this case, we also do not expect significant differences in terms of results between multiple dimension interpolation and multi-step interpolation.

*6) P6, l10: the interpolation error could be reduced by having finer LUT grids and / or use higher order interpolation (e.g. spline) without increasing unreasonably the LUT size. The author should justify why this was not preferred. When changing fitting window, in principle a new calculation of PCs should be performed. Is it the case or does the algorithm simply sample the original PCs over the new fitting range? If so, what is the impact of this simplified approach on the results?*

We agree that using finer LUT grids may reduce the interpolation error. We did attempt to use 50-DU $SO_2$ grid and found the interpolation error to be still fairly large at short wavelengths. So the estimate is that the $SO_2$ grid space needs to be 25 DU or finer. That translates into a 4-time increase in terms of LUT size, which is already 3.3 GB in total (again, we need Jacobians for the entire spectral range). The other factor that prevented us from using a finer $SO_2$ grid is the amount of RT calculations required to build these LUTs. Using 21 CPUs, the RT calculation for the current table took ~3 months. Increasing this by 4 times was deemed to be too time consuming.

We have also tested spline interpolation and found significant interpolation error for 100-DU grid space (see the figure below). A fine $SO_2$ grid may also help to reduce interpolation error, but would again significantly increase both the size of LUT and the time to build it.

[Figure]

Figure. Comparison between spline interpolated (red line) and directly calculated (blue line) $SO_2$ Jacobians indicates significant interpolation error at wavelengths < 315 nm.

As for the PCs, we take the "simplified" approach by sampling the original PCs in the new fitting range. In this approach, the PCs are treated as if they were reference spectra or cross sections in the DOAS fitting. To find out whether this simplification results in

large difference in retrievals, we also conducted a test in which PCs were generated each time a new fitting window was used. The TRU retrievals for orbit 21635 using these two different PCA approaches are given in the figure below. For the most part, the two approaches generate very similar results. One exception is that retrievals failed to detect $SO_2$ for one row near the southwest edge of the plume in (b) – likely due to $SO_2$ signatures in the new PCs that cause over-fitting. This, to us, points to a challenge in using new PCs for each different window. For a set of PCs from a fixed spectral range (i.e., the "simplified approach"), it is relatively easy to optimize the methods and criteria to exclude those PCs with $SO_2$ features in the spectral fitting. On the other hand, if we are to generate new PCs for each new window, the new criteria will need to be selected for each window to avoid over-fitting.

[Figure]

Figure. TRU $SO_2$ retrievals for OMI orbit 21635 using (a) PCs sampled from the original PCs and (b) PCs generated for each new fitting window.

*7) Ge et al., 2016 cannot be found in the references list.*

References are ordered alphabetically and Ge et al., 2016 can be found in the list after Flynn et al., 2014.

*8) Section 3.1 contains no new information compared to previous work (Li et al., 2013). The author should consider replacing this section by other results.*

We'd like keep this section given that it helps readers less familiar with the algorithm to understand the PCA retrieval approach, and also that we have added more detailed information to further illustrate the method, including an additional figure in the supplemental information (Figure S1) per the reviewer's suggestion.

*9) In section 3.2, it would be interesting to show the inverted reflectivity R and illustrate the wavelength range used (smallest wavelength).*

We have included the requested maps of R and the smallest wavelength of the fitting in the supplemental information (Figures S2 and S3).

*10) Section 3.2 and 3.3: it is not clear why in one case (Kasatochi) the LF and PCA STL results are so close (is PCA really improving the situation?) and for the other case*

*(Sierra Negra) the two algorithms produce results that are so different. Please expend the discussion.*

The sensitivity saturation issue for the LF algorithm is likely more serious for TRL and TRM retrievals than for STL retrievals, particularly for large $SO_2$ loading. As illustrated in the figure below, when $SO_2$ is greater than ~100 DU, the rate of N value increase with $SO_2$ is much smaller for the TRL and TRM profiles than for the STL profiles. So the LF algorithm probably handles the saturation issue relatively well for STL retrievals, but not for TRM and TRL retrievals.

As a result, the main issue for LF in the Kasatochi case is not with STL retrievals, but rather with TRM retrievals, which yields a total $SO_2$ mass only slightly greater than that from STL results. Even if one knows that the actual plume height is ~10 km and attempts to interpolate STL and TRM results, the resulting $SO_2$ mass will still be much smaller than PCA (which has much larger TRU and TRM retrievals) and other algorithms (as referenced in the manuscript).

As for the Sierra Negra case, the comparison is made for TRL retrievals. Given the expected severe saturation issue with LF, it is not surprising that LF algorithm significantly underestimates the overall $SO_2$ loading and the max $SO_2$ amount as compared with both PCA and ISF algorithms.

Overall, we feel that the improvement by PCA in both cases is evidenced by much better agreement with other algorithms/instruments. We have added some of this discussion to the revised manuscript.

[Figure]

Figure. Calculated N value at 315 nm for different $SO_2$ amounts (normalized to $SO_2 = 1$ DU), assuming the same other conditions ($O_3 = 325$ DU, SZA = 45°, VZA = 40°, RAZ = 95°, constant SLER = 0.05) but different $SO_2$ profiles.

*11) Section 3.2, Fig 5: it is surprising to see a strong change between PCA TRU and PCA STL. The expected change in measurement sensitivity from 13 to 18 km should be small and is incompatible with this observed change.*

Thank you for raising this question. We conducted forward radiative transfer calculations assuming typical observation conditions within the Kasatochi plume. The results below indicate ~25% difference in $SO_2$ Jacobians between STL and TRU profiles. The difference between STL and TRU retrievals will be even greater, since the TRU Jacobians can be even smaller with a higher initial estimate of $SO_2$ (the figure below assumes the same $SO_2$ amount for the two profiles). We have added the figure to the supplemental information (Figure S5) and some of the discussion to the revised manuscript.

[Figure]

Figure. $SO_2$ Jacobians calculated with VLIDORT for typical conditions within the Kasatochi volcanic plume ($O_3 = 325$ DU, $SO_2 = 250$ DU, SZA $= 35°$, VZA $= 50°$, RAZ $= 35°$, constant SLER $= 0.4$) assuming STL (black line) and TRU (red line) $SO_2$ profiles.

*12) P10, l19: the author should describe better what are those challenges.*

In the revised manuscript we have added discussion on some of the challenges in continuing OMI $SO_2$ data with OMPS, including two figures in the supplemental information (Figures S6 and S7).

*13) P12, l2: The author states that differences between OMI and OMPS SO2 VCDs are attributed to differences in spatial resolution but, based on Fig 8, it is hard to believe. It is clear that there are patterns (with higher SO2) in the SO2 map which are typically of the same size or larger than the OMPS pixels. Please clarify.*

To clarify this, we have added a figure in the supplementary information (Figure S8). The size of the OMPS pixel is much greater than OMI, and there are typically several OMI

pixels within each OMPS pixel. There is also substantial variability in OMI SO$_2$ within each OMPS pixel. For the peak part of the plume, OMI SO$_2$ varies from ~20 DU to nearly 70 DU within a single OMPS pixel. The maximum SO$_2$ retrieved from OMPS would therefore be smaller than OMI, even if the measurements had been made at the same time by the two instruments.

*Typos*
*-page 2, l20: "due the small .." –> "due to the small.."*

Corrected.

*-page 5, l7 and 19: the same notation (I) is used for two different quantities (according to the text): sun-normalized earthshine radiance and backscatter radiances at TOA. Please clarify.*

We have changed "sun-normalized earthshine radiance" to "Sun-normalized backscattered radiances at TOA".

---

## Author Comment (AC2) · 24 Dec 2016

Response to review of "*New-generation NASA Aura Ozone Monitoring Instrument (OMI) volcanic SO2 dataset: Algorithm description, initial results, and continuation with the Suomi- NPP Ozone Mapping and Profiler Suite (OMPS)*" (doi:10.5194/amt-2016-221).

Referees' comments in *Italic*, Responses in blue

**Anonymous Referee #1**
*Review of "New-generation NASA Aura Ozone Monitoring Instrument (OMI) volcanic SO2 dataset: Algorithm description, initial results, and continuation with the Suomi-NPP Ozone Mapping and Profiler Suite (OMPS) " by Li et al.*

*Li et al. present a variant on their very successful PCA boundary-layer SO2 retrieval algorithm, aimed here at retrieving volcanic SO2, and apply it to OMI and OMPS. For larger SO2 loading they utilize long wavelengths in their retrieval. The authors find greatly reduced retrieval noise, and removal of a high bias, with this product. Its successful application to OMPS will help ensure a continuation of the OMI volcanic SO2 data record. This is clear and well written and represents an advance in the retrieval of SO2 from UV/vis satellite spectra. I recommend it be published once the reviewers address the points given below:*

We thank the referee for the review and suggestions. Following these suggestions, we have made changes to the revised manuscript. Please see below our responses to the specific comments.

*Page 12, line 24: "In the absence of information on SO2 plume height . . ." - OMI should have information on plume height in its spectra, at least for larger eruptions. What about retrieving SO2 plume height? This was demonstrated previously for OMI by Yang et al. (2009). Presumably this knowledge would greatly reduce one of the larger sources of error for users. Please address this.*

*Yang, K., X. Liu, N. A. Krotkov, A. J. Krueger, and S. A. Carn (2009), Estimating the altitude of volcanic sulfur dioxide plumes from space borne hyper-spectral UV mea-surements, Geophys. Res. Lett., 36, L10803, doi:10.1029/2009GL038025.*

We agree that for large volcanic eruptions, OMI radiances may contain some information about the height of the volcanic plume as demonstrated by *Yang et al.* (2009). We also note that the plume height retrievals rely on shorter wavelengths (< 313 nm) and also require extensive on-line radiative transfer calculations, given the large potential interpolation error at these wavelengths. With the associated computational cost and execution time, the plume height retrievals are, for now, probably best done for case studies instead of as a part of an operational global product. We have added this discussion to the revised manuscript.

*Section 3: More detailed/quantitative/spatial comparisons should be made with GOME-2. E.g., Figure 5 and figure 7. GOME2 is mentioned in passing but real comparisons would provide additional confidence in this new product (different sensor + different*

*algorithm). Provide GOME2 VCD maps for one of the eruptions studied.*

Thank you for the suggestion. We have added a figure of GOME-2A SO$_2$ from the GDP (GOME Data Processors) retrievals from DLR in the supplemental information (Figure S4). In the revised manuscript, we have also added relevant discussion on the total SO$_2$ loading derived from the GOME-2A retrievals. We feel that detailed comparison between OMI and GOME-2A spatial distribution would be difficult, given the large differences in pixel size and overpass time.

*Page 5, line 11: change "computationally too expensive" to "too computationally expensive"*

Changed.